# Modeling Human Exploration Through Resource-Rational Reinforcement Learning

**Marcel Binz**
MPI for Biological Cybernetics
Tübingen, Germany
marcel.binz@tue.mpg.de

**Eric Schulz**
MPI for Biological Cybernetics
Tübingen, Germany
eric.schulz@tue.mpg.de

## Abstract

Equipping artificial agents with useful exploration mechanisms remains a challenge to this day. Humans, on the other hand, seem to manage the trade-off between exploration and exploitation effortlessly. In the present article, we put forward the hypothesis that they accomplish this by making optimal use of limited computational resources. We study this hypothesis by meta-learning reinforcement learning algorithms that sacrifice performance for a shorter description length (defined as the number of bits required to implement the given algorithm). The emerging class of models captures human exploration behavior better than previously considered approaches, such as Boltzmann exploration, upper confidence bound algorithms, and Thompson sampling. We additionally demonstrate that changing the description length in our class of models produces the intended effects: reducing description length captures the behavior of brain-lesioned patients while increasing it mirrors cognitive development during adolescence.

## 1 Introduction

Knowing how to efficiently balance between exploring unfamiliar parts of an environment and exploiting currently available knowledge is an essential ingredient of any intelligent organism. In theory, it is possible to obtain a Bayes-optimal solution to this exploration-exploitation dilemma by solving an augmented problem known as a Bayes-adaptive Markov decision process (BAMDP, Duff, 2003). However, BAMDPs are intractable to solve in general and analytical solutions are only available for a few special cases [Gittins, 1979]. The intractability of this optimal solution prompted researchers to develop a body of heuristic strategies [Auer et al., 2002, Kaufmann et al., 2012, Russo et al., 2017, Russo and Van Roy, 2014]. Most of these heuristics can be divided into two broad categories: directed and random exploration strategies [Wilson et al., 2014, Schulz and Gershman, 2019]. Directed exploration strategies provide bonus rewards that encourage the agent to visit parts of the environment that ought to be explored, whereas random exploration strategies inject some form of stochasticity into the policy.

Having access to a vast amount of existing exploration strategies leads to the question: which of them should we use when building artificial agents? To answer this question, we may take inspiration from how people approach the exploration-exploitation dilemma. Human exploration has been studied extensively in the multi-armed bandit setting [Mehlhorn et al., 2015, Wilson et al., 2021, Brändle et al., 2021]. Prior work indicates that people substantially deviate from the Bayes-optimal strategy even for the simplest bandit problems [Steyvers et al., 2009, Zhang and Angela, 2013, Binz and Endres, 2019]. They, however, use uncertainty estimates to intelligently guide their choices through a combination of both directed and random exploration [Wilson et al., 2014, Gershman, 2018]. The question of when and why individuals rely on a particular exploration strategy has been under-explored so far.

36th Conference on Neural Information Processing Systems (NeurIPS 2022).

We take the first steps towards answering these questions by looking at human exploration from a resource-rational perspective [Gershman et al., 2015, Lieder and Griffiths, 2020, Binz et al., 2022]. More specifically, we investigate the hypothesis that people solve the exploration-exploitation dilemma by making optimal use of limited computational resources. To test this hypothesis, we devise a family of resource-rational reinforcement learning algorithms by combining ideas from meta-learning [Bengio et al., 1991, Schmidhuber et al., 1996] and information theory [Hinton and Van Camp, 1993]. The resulting model – which we call resource-rational RL$^2$ (RR-RL$^2$) – implements a free-standing reinforcement learning algorithm that achieves optimal behavior subject to the constraint that it can be implemented with a given number of bits.

We demonstrate that RR-RL$^2$ captures many aspects of human exploration by reanalyzing data from three previously conducted psychological studies. First, we show that it explains human choices in a two-armed bandit task better than traditional approaches, such as Thompson sampling [Thompson, 1933], upper confidence bound (UCB) algorithms [Kaufmann et al., 2012], and mixtures thereof [Gershman, 2018]. We then verify that the manipulation of computational resources in our class of models matches the manipulation of resources in human subjects in two different contexts. Taken together, these results enrich our understanding of human exploration and provide insights into how to improve the exploratory capabilities of artificial agents.

## 2 Methods

We start by describing the general problem setting considered in this article and its optimal solution, followed by a brief summary of the meta-reinforcement learning framework. We then show how to augment the standard meta-reinforcement learning objective with an information-theoretic constraint, allowing us to construct reinforcement learning algorithms that trade-off performance against the number of bits required to implement them.

### 2.1 Notation and Preliminaries

Each task considered in this article can be interpreted as a multi-armed bandit problem. In a $k$-armed bandit problem, an agent repeatedly interacts with $k$ slot machines that are associated with a reward distribution $p(r_t|a_t, \omega)$ with unknown parameters $\omega$. In each time-step, the agent selects an action $a_t$ and is provided a reward $r_t$ based on the associated reward distribution.

The goal of an agent is to select actions such that the sum of rewards over a finite horizon $H$ is maximized. We assume that the agent additionally has access to a prior distribution $p(\omega)$ over bandit problems that it may encounter, which can be updated after observing a history of observations $h_t := (a_1, r_1, \ldots, a_{t-1}, r_{t-1})$ by applying Bayes' rule:

$$p(\omega|h_t) \propto p(\omega) \prod_{m=1}^{t-1} p(r_m|a_m, \omega) \tag{1}$$

The policy that optimally trades-off exploration and exploitation can be obtained by reasoning how the agent's beliefs about reward functions evolve with future observations [Martin, 1967]. Formally, this is accomplished by transforming the original bandit problem into a corresponding BAMDP defined by the tuple $(\mathcal{H}, \mathcal{A}, H, T, R)$. In this augmented problem, $\mathcal{H}$ represents the set of all possible histories, while $\mathcal{A}$ and $H$ correspond to the action space and the horizon of the original bandit problem. The transition probabilities $T$ and reward function $R$ are given by:

$$T(h_{t+1}|a_t, h_t) = p(r_t|a_t, h_t)\delta\left[h_{t+1} = (h_t, a_t, r_t)\right] \tag{2}$$
$$R(a_t, h_t) = \mathbb{E}_{p(r_t|a_t, h_t)}\left[r_t\right] \tag{3}$$

with the marginal reward probabilities:

$$p(r_t|a_t, h_t) = \int p(r_t|a_t, \omega)p(\omega|h_t)d\omega \tag{4}$$

The policy that maximizes the sum of rewards in the BAMDP corresponds to the Bayes-optimal policy for the original bandit problem.

## 2.2 Meta-Reinforcement Learning

While the BAMDP formalism provides a precise recipe for deriving a Bayes-optimal policy, finding an analytical expression of this policy is typically not possible. Recent work on meta-reinforcement learning, however, has shown that it is possible to learn an approximation to it [Wang et al., 2016, Ortega et al., 2019, Zintgraf et al., 2019]. Duan et al. [2016] refer to this approach as $\text{RL}^2$ because it uses a traditional reinforcement learning algorithm to meta-learn another free-standing reinforcement learning algorithm.

$\text{RL}^2$ parametrizes the to-be-learned reinforcement learning algorithm with a general-purpose function approximator. Typically, this function approximator takes the form of a recurrent neural network that receives the last action and reward as inputs, uses them to update its hidden state, and produces a policy that is conditioned on the new hidden state. Let $\mathbf{W}$ denote the parameters of this recurrent neural network. In an outer-loop meta-learning process, the network is then trained on the prior distribution over bandit problems $p(\omega)$ to find the history-dependent policy $\pi(a_t|h_t, \mathbf{W})$ that maximizes the sum of obtained rewards. If the meta-learning procedure has successfully converged to its optimum, $\text{RL}^2$ implements a free-standing reinforcement learning algorithm that mimics the Bayes-optimal policy. Importantly, learning at this stage is fully implemented through the forward dynamics of the recurrent neural network and no further updating of its parameters is required.

## 2.3 Resource-Rational $\text{RL}^2$

We transform $\text{RL}^2$ into a resource-rational algorithm by augmenting its meta-learning objective with an information-theoretic constraint and refer to this resource-rational variant as $\text{RR-RL}^2$. More precisely, $\text{RR-RL}^2$ is obtained by solving the following optimization problem:

$$\max_{\mathbf{\Lambda}} \; \mathbb{E}_{q(\mathbf{W}|\mathbf{\Lambda})p(\omega)\prod p(r_t|a_t,\omega)\pi(a_t|h_t,\mathbf{W})} \left[ \sum_{t=1}^{H} r_t \right]$$

$$\text{s.t. KL}\left[q(\mathbf{W}|\mathbf{\Lambda})||p(\mathbf{W})\right] \leq C \tag{5}$$

The first component of Equation 5 corresponds to the standard meta-reinforcement learning objective, while the second component ensures that the Kullback–Leibler (KL) divergence between a stochastic parameter encoding $q(\mathbf{W}|\mathbf{\Lambda})$ and a prior $p(\mathbf{W})$ remains smaller than some constant $C$. The KL term can be interpreted as the description length of neural network parameters, i.e., the number of bits required to store them.[1] $\text{RR-RL}^2$ therefore optimally trades-off performance against the number of bits required to implement the emerging reinforcement learning algorithm. Note that the objective from Equation 5 can also be motivated by a PAC-Bayes bound on generalization performance to unseen tasks [Yin et al., 2019, Rothfuss et al., 2020, Jose and Simeone, 2020]. While we focus on the resource-rational interpretation in the present article, we believe that both of these perspectives are complementary.

In practice, we solve a sample-based approximation of the optimization problem in Equation 5 using a standard on-policy actor-critic algorithm [Mnih et al., 2016, Wu et al., 2017]. We rely on a dual gradient ascent procedure [Haarnoja et al., 2018] to ensure that the constraint is satisfied. Appendix A contains a complete description of the network architecture, choices of prior and encoding distribution, and the meta-learning procedure. A public implementation of our model and the following analyses is available under `https://github.com/marcelbinz/resource-rational-reinforcement-learning`.

Which exploration strategies $\text{RR-RL}^2$ implements will partially depend on its available computational resources. Models with limited resources must implement strategies that rely on simple computations, such as computing average rewards or noisy estimates thereof. Models with access to more resources, on the other hand, can spend some of them to compute more complex statistics, such as uncertainty estimates, and incorporate these into their decision-making process. Moreover, what strategies are resource-rational not only depends on the computational resources of the decision-maker but also on the characteristics of the particular problem under consideration. Problems with a short task horizon, for instance, require less exploration and thereby make exploitation more appealing, while problems with a longer horizon allow for the application of more sophisticated exploration strategies.

---

[1]The desired coding length can, for example, be achieved using bits-back coding [Hinton and Van Camp, 1993] or minimal random coding [Havasi et al., 2018].

**(a) Exploration Strategies**

**(b) Embedding**

Figure 1: Illustration of exploration strategies implemented by RR-RL$^2$ (also see Equation 6 and the corresponding explanation in the main text). (a) Probit regression coefficients obtained by fitting the hybrid model to data simulated by RR-RL$^2$ with varying description lengths (depicted on a logarithmic scale). The blue line shows the influence of the estimated mean on choices (Boltzmann exploration), the orange line shows the influence of the option's uncertainty estimates (UCB-based exploration), and the green line shows the influence of the uncertainty-scaled estimated mean (which corresponds to Thompson sampling in this particular task). (b) UMAP embeddings of probit regression coefficients for RR-RL$^2$ and human participants.

## 3 Modeling Human Exploration

We now demonstrate that RR-RL$^2$ explains human choices on both a qualitative and quantitative level. We first show that varying its description length leads to a set of diverse exploration strategies, allowing us to capture individual differences in human decision-making. When reanalyzing data from a two-armed bandit benchmark [Gershman, 2018], we furthermore find that RR-RL$^2$ beats previously considered algorithms in terms of fitting human behavior by a large margin.

**Experimental Design:** The behavioral data-set of Gershman [2018] contains records of 44 participants who each played 20 two-armed bandit problems with an episode length of $H = 10$. The mean reward for each arm $a$ was drawn from $p(\omega_a) = \mathcal{N}(0, 10)$ at the beginning of the task and the reward in each time-step from $p(r_t|a_t, \omega) = \mathcal{N}(\omega_{a_t}, 1)$.

**Analysis:** To analyze the set of emerging exploration strategies, we adopted a method proposed by Gershman [2018]. He assumed that an agent uses Bayes' rule as described in Equation 1 to update its beliefs over unobserved parameters. If prior and reward are both normally distributed, the posterior will also be normally distributed and the corresponding updating rule is given by the Kalman filtering equations. Let $p(\omega_a|h_t) = \mathcal{N}(\mu_{a,t}, \sigma_{a,t})$ be the posterior distribution at time-step $t$. Based on the parameters of this posterior distribution, he then defined the following probit regression model:

$$p(A_t = 0|h_t, \mathbf{w}) = \mathbf{\Phi}\left(\mathbf{w}_1 \mathbf{V}_t + \mathbf{w}_2 \mathrm{RU}_t + \mathbf{w}_3 \frac{\mathbf{V}_t}{\mathrm{TU}_t}\right) \tag{6}$$

$$\mathbf{V}_t = \mu_{0,t} - \mu_{1,t}$$
$$\mathrm{RU}_t = \sigma_{0,t} - \sigma_{1,t}$$
$$\mathrm{TU}_t = \sqrt{\sigma_{0,t}^2 + \sigma_{1,t}^2}$$

with $\mathbf{\Phi}$ denoting the cumulative distribution function of a standard normal distribution. Equation 6 is also referred to as the hybrid model as it contains several known exploration strategies as special cases. We can recover a Boltzmann-like exploration strategy for $\mathbf{w} = [\mathbf{w}_1, 0, 0]$, a variant of the UCB algorithm for $\mathbf{w} = [\mathbf{w}_1, \mathbf{w}_2, 0]$, and Thompson sampling for $\mathbf{w} = [0, 0, 1]$.

Fitting the coefficients of the hybrid model to behavioral data allows us to inspect how much a given agent relied on a particular exploration strategy. Previously, Gershman [2018] has applied this form of analysis to human data, which revealed that people rely on a mixture of directed and random exploration. In this article, we extend this approach to artificial data generated by RR-RL$^2$.

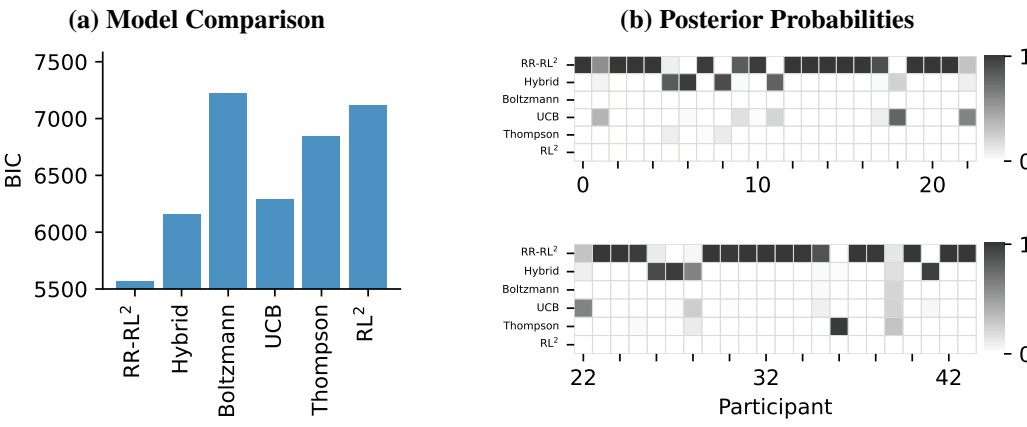

Figure 2: Model comparison results on the two-armed bandit data from Gershman [2018]. (a) Bayesian information criterion (BIC) values for the aggregated data of all participants. Lower values correspond to a better fit to human behavior. (b) Posterior probabilities for each model and participant. Higher values correspond to a better fit to human behavior.

**Results:** We trained RR-RL$^2$ with a targeted description length of $\{1, 2, \dots, 10000\}$ nats on the same distribution used in the original experimental study and examined how the description length of these models influences their exploration behavior using the previously described probit regression analysis. Figure 1 (a) illustrates the results of this analysis. We find that RR-RL$^2$ implements a Boltzmann-like exploration strategy for description lengths between 1 and 100 nats. Note that behavior at this stage is quite noisy as indicated by the small probit regression weights (average regression coefficient of 0.15). Beginning from 100 nats, we observe a rise of the factor corresponding to Thompson sampling, which continues to rise until the limit of 10000 nats. Between 100 and 1000 nats, we additionally find minor influences of a Boltzmann-like exploration strategy (average regression coefficient of 0.19). For a description length of 1000 nats and larger, Boltzmann-like exploration diminishes and is replaced with minor influences of a UCB-based strategy (average regression coefficient of 0.23).

Having established that different styles of exploration emerge in RR-RL$^2$ depending on its description length, we next set out to test how well it explains human choices. In order to do so, we conducted a Bayesian model comparison [Bishop, 2006]. A detailed summary of our comparison procedure is provided in Appendix B. We used the Bayesian information criterion (BIC, Schwarz, 1978) as an approximation to the model evidence. The resulting BIC values for each candidate model are illustrated in Figure 2 (a). We find that the BIC value for RR-RL$^2$ is substantially lower compared to that of the hybrid model (5562.63 against 6158.91) when aggregated across all participants; all other models provide a less adequate fit to human choices. The majority of participants ($n = 32$) was best described by RR-RL$^2$ and the protected exceedance probability (PXP), which measures the probability that a particular model is the most frequent within a set of alternatives [Rigoux et al., 2014], also favored RR-RL$^2$ decisively (PXP $\approx 1$). We provide a detailed illustration of the posterior probabilities for each model and participant in Figure 2 (b).

Finally, we compared the probit regression coefficients of human participants to the ones of RR-RL$^2$. Figure 1 (b) shows a two-dimensional UMAP embedding [McInnes et al., 2018] of these coefficients. The figure reveals a set of diverse exploration strategies within the human population and confirms that RR-RL$^2$ captures the overall variability in human exploration appropriately.

## 4 Manipulating Computational Resources

RR-RL$^2$ also makes precise predictions about what should happen if computational resources are actively manipulated. Do these predictions align with the actual behavior of people? Providing answers to this question is non-trivial because we cannot simply ask a person to use an algorithm with a shorter or longer description length. There are, however, two types of studies that can provide insights. The first type consists of lesion studies that compare the behavior of healthy participants to that of participants suffering from brain damage, whereas the second type consists of developmental

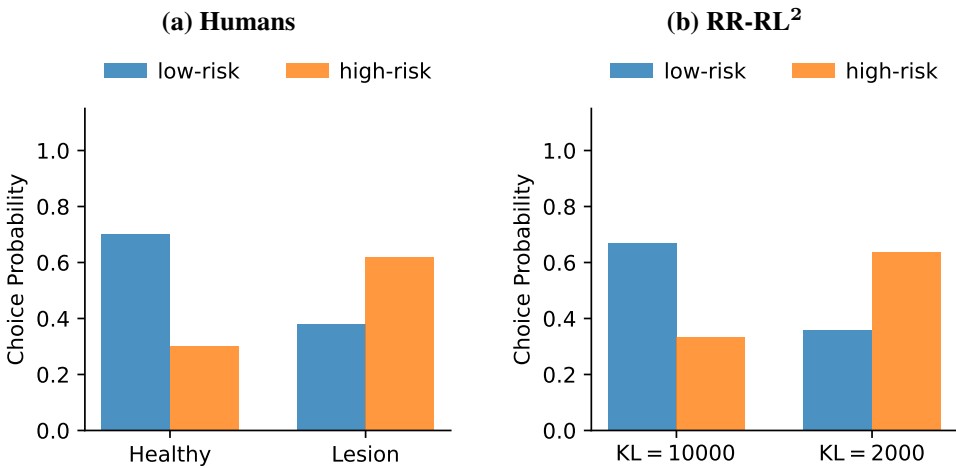

Figure 3: Probability of selecting low- and high-risks arms in the Iowa Gambling Task. (a) Human data taken from Bechara et al. [1994]. The probability of selecting an inferior high-risk arm is increased in participants with vmPFC damage. (b) Data simulated from RR-RL$^2$ with large and small description length. The probability of selecting an inferior high-risk arm is increased for models with fewer bits, mirroring the results of the original study.

studies that investigate how behavior evolves during cognitive development. We next take a look at an example of each of them and demonstrate that RR-RL$^2$ reproduces their key findings.

## 4.1 Damage to Ventromedial Prefrontal Cortex

There has been a long history of analyzing people with brain lesions in cognitive neuroscience [Damasio and Damasio, 1989]. We focused on a particular study conducted by Bechara et al. [1994] for the purpose of this article and predicted that reducing the description length of RR-RL$^2$ should correspond to the behavior of brain-lesioned patients.

**Experimental Design:** To probe decision-making in clinical populations, Bechara et al. [1994] introduced an experimental paradigm called the Iowa Gambling Task (IGT). The IGT involves 100 choices in a single four-armed bandit problem. Two of the arms are high-risk arms, while the other two are low-risk arms. High-risk arms result in a constant positive reward of 100 but have a chance to yield a noisy penalty with an expected value of 125. Low-risk arms result in a constant positive reward of 50 but have a chance to yield a noisy penalty with an expected value of 25. A complete list of trials is printed in Table D1. Bechara et al. [1994] used the IGT to compare the decision-making of healthy participants to that of participants with ventromedial prefrontal cortex (vmPFC) damage.

**Analysis:** The focus of our analysis was the proportion of selected low- and high-risk arms across the entire experiment. High-risk arms cause an average loss of 25 points per trial, while low-risk arms provide an average payoff of 25 points per trial. We should therefore expect an agent to select the superior low-risk arms with higher frequency. Healthy participants are indeed able to learn about the structure of the task and will after a while start to sample the superior low-risk arms. Participants with vmPFC damage, however, continue to sample to the inferior high-risk as illustrated in Figure 3 (a). This pattern is striking because performance in these subjects remains worse than chance regardless of how often they interact with the task.

**Results:** RR-RL$^2$ requires a distribution over bandit problems for meta-learning, but participants in the IGT only encountered a single bandit task. Therefore, we cannot directly use the task of the original study for meta-learning as we have done in the previous example. We instead constructed a distribution over bandit problems that maintains the key characteristics of the IGT:

- The positive reward component was independently sampled for each arm from a uniform distribution with a minimum value of 0 and a maximum value of 150.
- The mean across all trials of the negative reward component was also sampled from a uniform distribution with a minimum value of 0 and a maximum value of 150.

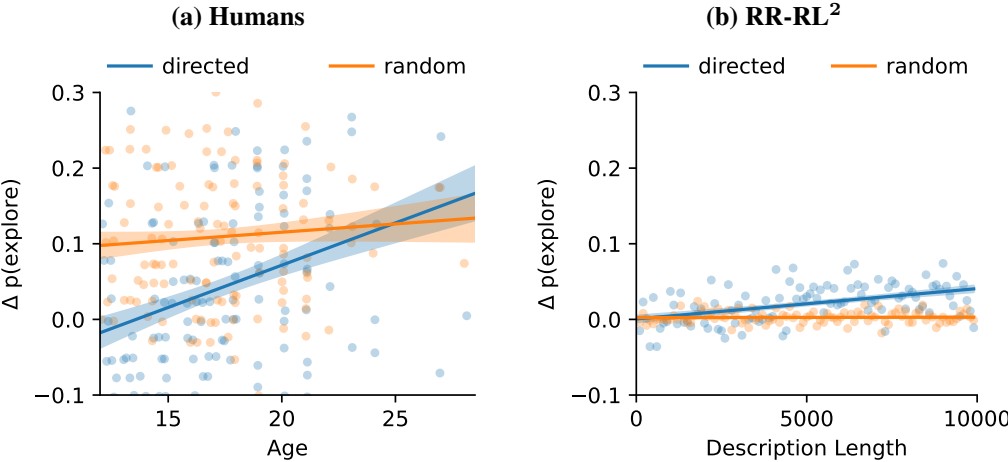

Figure 4: Illustration of strategic directed and random exploration in the horizon task. (a) Human data from Somerville et al. [2017]. During adolescence, people start to engage more in strategic directed exploration, whereas strategic random exploration remains constant over time. (b) Data simulated from RR-RL$^2$ with varying description lengths. Like in the human data, we observe an increase in strategic directed exploration, but no change in strategic random exploration.

- The negative reward component had an occurrence probability sampled randomly from a uniform distribution with a minimum value of $0.05$ and a maximum value of $0.95$.

- We furthermore added additive noise sampled from a zero-mean normal distribution with a standard deviation of $10$ to the negative reward component in each time-step.

We trained RR-RL$^2$ with a targeted description length of $\{100, 200, \ldots, 10000\}$ nats on the previously described distribution. When tested on the IGT, we find that RR-RL$^2$ replicates the pattern reported by Bechara et al. [1994]. Models with a high description length successfully solve the task by selecting low-risk arms in the majority of time-steps. If description length is however sufficiently reduced, RR-RL$^2$ predominately samples high-risk arms. We illustrate this behavior for two example models in Figure 3 (b). Figure D2 provides a more detailed picture of how description length mediates choice behavior.

In summary, our analysis sheds light on why brain-lesioned patients display below-chance performance in the IGT. Intuitively, any resource-limited agent must primarily devote its computational resources to things that are easy to estimate. In the IGT, the deterministic positive reward component is easier to estimate than the noisy negative component. An agent with significantly restricted resources will thus focus on the positive component while ignoring the negative. In turn, the agent will assign higher estimated payoffs to the inferior high-risk arms and therefore select them more frequently. We found that RR-RL$^2$ implements this behavior and that reducing its description length captured participants with lesioned vmPFC.

### 4.2 Developmental Trajectories

People are not born with fully-developed cognitive abilities but instead develop them during their lifetime. In this section, we tested whether increasing the description length of RR-RL$^2$ matches the behavioral trajectories of people as they grow up. To test this hypothesis, we reanalyzed data collected by Somerville et al. [2017], who studied changes in exploration behavior between early adolescence and adulthood.

**Experimental Design:** In their study, Somerville et al. [2017] made use of an experimental paradigm known as the horizon task [Wilson et al., 2014]. Each task was based on a two-armed bandit problem and involved four forced-choice trials, followed by either one or six free-choice trials. Participants were aware of the number of remaining choices and could use this information to guide their behavior. The mean reward of one of the arms was drawn randomly from $\{40, 60\}$, while the mean reward for the other was determined by sampling the difference to the first arm from $\{4, 8, 12, 20, 30\}$. The

arrangement of arms as well as the sign of their difference was randomized. In each time-step, the observed reward was sampled from a normal distribution with the corresponding mean value and a standard deviation of 8. The addition of forced-choice trials allowed to control the amount of information that was available to participants. They either provided an equal amount of information for both arms (i.e., two observations each) or an unequal amount of information (i.e., a single observation from one arm, three from the other). In total, Somerville et al. [2017] collected data for 147 participants between the ages of 12.08 and 28, completing 160 bandit tasks each.

**Analysis:** Following Somerville et al. [2017], we used the decision in the first free-choice trial to distinguish between different types of exploration. In the unequal information condition, a choice was classified as directed exploration if it corresponded to the option that was observed fewer times during the forced-choice trials. In the equal information condition, a choice was classified as random exploration if it corresponded to the option with the lower estimated mean. We refer to an exploration behavior as strategic if it occurs more frequently in the long horizon tasks compared to the short horizon tasks. Somerville et al. [2017] found – as shown in Figure 4 (a) – that strategic directed exploration emerges during adolescence, whereas strategic random exploration is age-invariant.

To quantify these effects, they fitted two independent linear regression models, using the probability of engaging in directed and random exploration as dependent variables. Both models used age, the corresponding horizon, and the interaction between the two as regressors. They found a significant effect of horizon in both conditions, indicating that participants engaged more in both directed and random exploration in tasks with a longer horizon. Furthermore, they found a significant interaction effect between horizon and age for directed exploration but not for random exploration, confirming that strategic directed exploration increases during cognitive development, while strategic random exploration remains constant over time.

**Results:** We trained RR-RL$^2$ with a targeted description length of $\{100, 200, \dots, 10000\}$ nats on the same distribution used in the original experimental study. Figure 4 (b) visualizes how strategic directed and random exploration change as the description length of RR-RL$^2$ increases. Matching the main result of the experimental study, we find that strategic directed exploration increases with description length, while strategic random exploration remains unaffected. We repeated the previously described regression analysis on data simulated by RR-RL$^2$ to quantify this conclusion (replacing age as a regressor with description length). The outcome of this analysis mirrored the results of the original study. We found a significant effect of horizon on both directed ($F_{1,194} = 56.50, p < 0.001, \eta^2 = 0.20$) and random exploration ($F_{1,194} = 6.80, p = 0.01, \eta^2 = 0.03$). This means that RR-RL$^2$ made more exploratory decisions of both types if it was beneficial to do so. We also found a significant interaction effect between horizon and description length on directed exploration ($F_{1,194} = 17.48, p < 0.001, \eta^2 = 0.06$) but not on random exploration ($F_{1,194} = 0.02, p = 0.89$). These results confirm that description length and age have comparable qualitative effects on the development of strategic exploration.

However, when comparing the effect sizes of our analysis to those from the experimental study, we find that the interaction effect between description length and horizon on directed exploration only amounts to around half of the effect between age and horizon. We speculated that part of this difference comes from a mismatch between the distribution used to train our models and what kind of tasks people expect in the experiment. People, for instance, might assume that task rewards are noisier than they are, which would require more exploratory choices, and, in turn, lead to stronger effects. We tested this hypothesis by retraining RR-RL$^2$ on the same distribution but with the standard deviation of the reward noise increased by 50%. While this modification increased the effect size of the interaction effect on directed exploration, it did not close the gap entirely, suggesting that there are additional – currently undiscovered – factors that contribute to the development of strategic directed exploration during adolescence. Furthermore, we found that, while strategic random exploration was unaffected by both age and description length, humans had a higher base rate of engaging in random exploration than our models. We, however, believe that this issue could be resolved by adding an $\varepsilon$-greedy error model to our models as doing so would naturally increase the base rate of random exploration.

# 5 General Discussion

The exploration-exploitation dilemma is one of the core challenges in reinforcement learning. How do humans arbitrate between exploration and exploitation, and which kind of exploration strategies do they engage in? We have put forward the hypothesis that people tackle this problem in a resource-rational manner. To test this hypothesis, we proposed a method for meta-learning reinforcement learning algorithms with limited description length. The resulting class of models – which we coined RR-RL$^2$ – makes precise predictions about how people make decisions. We have put these predictions to a rigorous test by comparing our model to data from three psychological studies. RR-RL$^2$ displayed key elements of human decision-making in all three of them:

1. It captured human exploration in a two-armed bandit task on both a qualitative and quantitative level.
2. Reducing its description length aligned with decision-making in brain-lesioned patients.
3. Increasing its description length reflected changes in exploration behavior attributed cognitive development.

In summary, our results demonstrate that it is possible to meta-learn resource-rational reinforcement learning algorithms and that human exploration is well-characterized by these very algorithms.

## 5.1 Limitations and Future Work

We have focused on comparing RR-RL$^2$ to human exploration in the simple multi-armed bandit setting. In the real world, however, people face much more sophisticated challenges that call for a richer repertoire of exploration strategies [Schulz et al., 2019, Brändle et al., 2021]. This criticism is not necessarily a shortcoming of the proposed model, which could in principle be applied to more complex tasks, but rather one regarding the experimental research in cognitive psychology, which has predominately focused on multi-armed bandit problems. In future work, we intend to develop new experimental paradigms that allow us to compare RR-RL$^2$ against human behavior in more complex settings. Potential paradigms may include contextual bandits, grid world problems, and multi-agent scenarios.

RR-RL$^2$ also places a constraint on a particular type of computational resource: the description length of the reinforcement learning algorithm in use. People, on the other hand, are subject to a variety of additional computational constraints. They can, for instance, only run algorithms with finite computation time or only store a restricted amount of chunks in their short-term memory [Collins et al., 2014]. Future work should aim to unify all of these constraints in a common framework.

It would also be interesting to investigate how RR-RL$^2$ could be implemented in a biologically plausible way. We believe that replacing the standard recurrent neural networks used in this article with spiking neural networks could provide one promising path towards this goal. Earlier work of Bellec et al. [2018] showed that it is indeed possible to train spiking neural networks in a meta-learning setting and would therefore provide an excellent starting point for this endeavor.

Finally, there are many alternative ways of how one could apply meta-learning to our setting. It is, for example, possible to meta-learn hyperparameters of standard reinforcement learning algorithms [Doya, 2002, Luksys et al., 2009], or to use model-agnostic meta-learning to find optimal weight initializations for neural networks that are trained by gradient descent [Finn et al., 2017]. Future work should compare these and other models against each other. For this to be effective, we believe that a set of commonly-accepted benchmarks needs to be established.

## 5.2 Conclusion

Many applications could benefit from the availability of human-like agents. Having access to such agents may be especially valuable in cooperative self-play scenarios, where training with them is crucial for successful coordination with people [Carroll et al., 2019, Strouse et al., 2021]. The traditional path towards constructing agents that learn and think like people is to take inspiration from the cognitive processes of the human mind and incorporate them into existing systems [Lake et al., 2017]. In this article, we have pursued a different approach. Instead of hard-coding cognitive processes directly into our agents, we have identified two computational principles – meta-learning

and resource rationality – that give rise to many aspects of human behavior. The presented approach is very general, easy to adapt to new domains, and can be scaled to more complex problem settings without major modifications.

Finally, we want to emphasize that low description lengths might not only be a biological necessity, but also a feature [Gigerenzer et al., 1999, Zador, 2019]. Implementing an algorithm in just a few bits acts as a strong form of regularization and could, in turn, produce exploration strategies that are applicable across domains. Hence, we believe that constructing artificial systems with such constraints could lead us towards more generally capable agents.

## Acknowledgments and Disclosure of Funding

This work was funded by the Max Planck Society, the Volkswagen Foundation, as well as the Deutsche Forschungsgemeinschaft (DFG, German Research Foundation) under Germany's Excellence Strategy–EXC2064/1–390727645.

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
