# OpenReview forum: "Modeling Human Exploration Through Resource-Rational Reinforcement Learning"
_NeurIPS.cc/2022/Conference — NeurIPS 2022 Accept_

### Official Review · Reviewer_m1ih · 2022-07-04

**Rating:** 9
**Confidence:** 5
**Soundness:** 4 excellent
**Presentation:** 4 excellent
**Contribution:** 4 excellent

**Summary:**

This paper develops a model called "RL3" which learns a free-standing RL algorithm by optimizing a resource-constrained objective function. The key contribution is to show how human-like patterns of exploration arise from the optimized algorithm. Using data from a two-armed Gaussian bandit task, they show that the learned algorithm quantitatively matches human behavioral data, and fits the data better than a number of previously proposed models. Importantly, the learned algorithm produces a mixture of directed, random, and Boltzmann sampling similar to what people produce in the task. Another interesting observation is that the mix of directed/random/Boltzmann changes with the capacity limit. The authors then show that reducing the capacity limit produces behavior similar to what has been observed in ventromedial prefrontal cortex patients. Finally the authors show that they can reproduce the developmental trajectory of directed vs. random exploration.

**Questions:**

- I understand that space is limited so I don't expect the authors to address this, but I think it would be very interesting to apply this approach to contextual bandits.
- I'm also curious about whether the implication here is that the brain contains many pretrained RL algorithms for different tasks, or a few that generalize across tasks.

**Limitations:**

I don't see any potential negative impacts. The authors devote a section of their discussion to limitations, though it's somewhat vague. I would have liked to see more specific predictions for future experimental work.

**Strengths And Weaknesses:**

Strengths:
- Overall, I really liked this paper. it presents an interesting and novel idea, the writing is clear, and the analysis is rigorous.
- The paper is quite rich, addressing data from healthy humans, lesion patients, and developmental trajectories.
- Not only can the model reproduce aspects of human exploration behavior, but the theory sheds light on which phenomena should be observed to emerge at different capacity limits (this should be testable).

Weaknesses:
- The meta-learning setup for the IGT is somewhat contrived. It might have made more sense to use tasks where there is a clearer distribution for meta-learning.

---

> ### Author Response · Authors · 2022-08-01
> **Reviewer 4**
>
> We thank the reviewer for the very positive feedback. We discuss the three minor points mentioned by the reviewer below. Note that we have changed RL$^3$’s name to RR-RL$^2$ based on another reviewer’s request.
>
> > The meta-learning setup for the IGT is somewhat contrived. It might have made more sense to use tasks where there is a clearer distribution for meta-learning.
>
> The reviewer definitely has a point here. But, in general, we think that the specific details about our constructed distribution are not too important and that our results would hold for many other meta-learning distributions as long as they share the key characteristics with the IGT: a noisy negative reward component (which is hard to estimate) and a deterministic positive reward component (which is easy to estimate).
>
> > I understand that space is limited so I don't expect the authors to address this, but I think it would be very interesting to apply this approach to contextual bandits.
>
> Testing our model on contextual bandits is certainly interesting and almost straightforward. The page limit prevents us from doing this in the current article, but we have added the following sentence to our newly uploaded version to discuss this possibility:
>
> In future work, we intend to develop new experimental paradigms that allow us to compare RR-RL$^2$ against human behavior in more complex settings. **Potential paradigms may include contextual bandits, grid world problems, and multi-agent scenarios.**
>
> > I'm also curious about whether the implication here is that the brain contains many pretrained RL algorithms for different tasks, or a few that generalize across tasks.
>
> This is a fascinating question, but we were not sure whether our paper can provide answers to it. To provide a better response to this question, we would need a more detailed explanation of what the reviewer has in mind.

---

> > ### Comment · Reviewer_m1ih · 2022-08-06
> > **Brain implications**
> >
> > My question about implications for the brain was directed at the idea that the brain stores amortized algorithms for exploration: are these stored separately for different task distributions, or is there one big task distribution? In any case, I think this is rather idle speculation and given space limitations it's okay with me if the authors don't go into it.

---

> > > ### Author Response · Authors · 2022-08-06
> > > **Response: Brain implications**
> > >
> > > Thanks for this clarification. We decided to train separate algorithms for each task distribution in the present paper. This choice was mainly made to simplify our evaluation. In reality, we think it is more plausible that there is one big task distribution and that the agent decides which strategy to execute based on some description of the encountered task (which could, for example, be given in the form of natural language). This setup would then also allow the agent to amortize components of exploration strategies between different tasks if it is beneficial to do so.

---

### Official Review · Reviewer_6vWe · 2022-07-11

**Rating:** 7
**Confidence:** 4
**Soundness:** 3 good
**Presentation:** 3 good
**Contribution:** 3 good

**Summary:**

This paper presents RL^3: a combination of meta reinforcement learning with information bottlenecks. RL^3 explains human exploration behavior as resource rational -- humans rely on limited description lengths (defined here as the amount of information in the weights of a reinforcement learning algorithm parameterized by a neural network) to balance exploration with exploitation in various multi-armed bandit tasks. By reanalyzing psychological data with both healthy adults, as well as clinical populations and adolescents, the paper shows that exploration behavior can be better explained using these resource rational principles as opposed to classic, hard-coded exploration strategies (such as Thompson sampling, for example).

**Questions:**

* While reasonable, this is a rather limited view of both description length, as well as requiring fairly strong assumptions about the form of the decision-making process underlying explore/exploit. For example, the presented method could not obviously extend to consider object-oriented or model-based exploration, which is likely critical for more ecological/complex tasks than bandit settings. Indeed, much of the work in AI for exploration has focused on these concepts. How do the authors see this method as being applicable to these more complex settings?
* The fit of RL^3 to both healthy adults and the lesioned population is very good, but the fit to the adolescent data is significantly worse. Both the effect size, but also the base rate of exploration, seem to be dramatically different from the human groups. Could the authors speculate on why this fit is not great, and whether there are changes to the model that could be made to improve the fit?
* One major missing piece for me was some intuition on how classic exploration strategies come out of the RL^3 approach. In particular, why is Thompson sampling seemingly the optimal strategy for exploration when a sufficiently high description length is allowed? And why does it require a high description length in the first place? Adding some intuition for this to the paper would help – especially in the first experiment for discussing Figure 1.


**Limitations:**

Limitations are adequately addressed by the authors. No ethical limitations to consider, but limitations of the approach are reasonably well described (although see my other comments for other potential important limitations).

**Strengths And Weaknesses:**

Strengths
* (Significance) This paper presents a compelling argument for how human exploration is more complex than simply exploiting or exploring. Cognitive scientists in particular will be inspired by this work to think about how to both move towards more complex exploration tasks, and in considering how to further use meta-learning techniques to model richer kinds of cognitive phenomena
* (Originality) Resource rationality itself is not new, but this is a very nice, straightforward application of the idea to demonstrate the flexibilty and variability of human exploration behavior.
* (Quality) The method is able to capture not just one experiment, but three separate sets of experiments separated by decades, and across exceptionally diverse populations (from lesioned groups, to healthy adults, to adolescents) without requiring fine-tuning or separate hyperparameters.
* (Clarity) The paper is well presented. The motivation is clear (with some minor exceptions outlined below), the method is well described, the results are fairly interpreted, and the conclusions are well laid out given the results.

Weaknesses
* (Significance) While the paper attempts to provide motivation for why these findings may be important to the ML/AI community, I am not entirely convinced of their arguments. In particular, the presented method does not seem obviously applicable to more complex settings that are commonly explored in ML/AI, often involving objects (like in videogames) and models. I think this contribution is important for computational cognitive science, but I am not sure it will have a big impact on the ML/AI audience.
* (Clarity) The paper title is both grandiose and also exceptionally vague. I would ask the authors to choose a paper title that is more indicative of the content of the paper. Key words like “resource rational” and “meta-learning” are some examples that I think could be used to make a more informative title.
* (Clarity) The abstract introduces terms that are not defined until the second page, specifically “description length”. Without context, description length could refer to an exceptionally large number of things, and is therefore quite confusing when mentioned in isolation. I would recommend changing the abstract such that description length is not mentioned, or is mentioned and unpacked in a clause or a single sentence to give the reader an understanding of what this specifically refers to.
* (Clarity) RL^3 seems like a poor method name given that RL^2 was so named since it was using reinforcement learning to train a reinforcement learning algorithm. Since this approach is not doing an extra meta-level of optimization, and is instead limiting the information in the weights of the inner RL algorithm, RL^3 is a very poor choice of name. What about RR-RL^2 for resource rational RL^2? Or IB-RL^2 for information bottleneck RL^2?
* (Clarity) Figure 1a is somewhat difficult to interpet as a standalone figure. I think I understand it as behaving more like Thompson sampling as the description length is increased, but is that what’s actually happening? Some extra description in the figure caption would really help with the interpretation. (See also my question in the “questions” section below).
* (Clarity) Figure 2a - is RL^3 really at 0 BIC? Is that theoretically possible? Or is the delta BIC with respect to the RL^3 model? If the latter, the caption should be updated accordingly.
* (Quality) Experimental results on development are much less compelling than for the other two experiments. The model is clearly doing something distinctly different from humans in that its base probability of exploration is 0 rather than a constant value. While the authors address some reasoning for why the effect size is lower for the model, there is no discussion of the differences in why the overall likelihood of exploration for people is so much higher. Could the authors comment on this?

---

> ### Author Response · Authors · 2022-08-01
> **Reviewer 3 (Part 1)**
>
> We are happy to hear that the reviewer found our paper compelling and clear. We appreciate the provided feedback and believe that incorporating it substantially improved the quality of our paper. We have addressed the reviewer’s concerns regarding significance, clarity, and quality as described below. Minor changes are already added to our newly uploaded version. For larger text changes, we describe what we will add to our camera-ready version (the camera-ready version allows for one additional page, but the rebuttal does not).
>
> >(Significance) While the paper attempts to provide motivation for why these findings may be important to the ML/AI community, I am not entirely convinced of their arguments. In particular, the presented method does not seem obviously applicable to more complex settings that are commonly explored in ML/AI, often involving objects (like in videogames) and models. I think this contribution is important for computational cognitive science, but I am not sure it will have a big impact on the ML/AI audience.
>
> Like the reviewer, we envision that this paper is mainly targeted at a computational cognitive science audience. Yet, we also have written in a way that a wider ML/AI audience is able to follow and potentially get inspired by it. NeurIPS historically also publishes cognitive science-focused work, and thus we think that putting the focus on this aspect is not an issue.
>
> >(Clarity) The paper title is both grandiose and also exceptionally vague. I would ask the authors to choose a paper title that is more indicative of the content of the paper. Key words like “resource rational” and “meta-learning” are some examples that I think could be used to make a more informative title.
>
> We would be open to changing our title to “Modeling human exploration through resource-rational reinforcement learning”, should this be permitted by the conference guidelines.
>
> >(Clarity) The abstract introduces terms that are not defined until the second page, specifically “description length”. Without context, description length could refer to an exceptionally large number of things, and is therefore quite confusing when mentioned in isolation. I would recommend changing the abstract such that description length is not mentioned, or is mentioned and unpacked in a clause or a single sentence to give the reader an understanding of what this specifically refers to.
>
> We thank the reviewer for spotting this issue and have adjusted our abstract as follows to fix it:
>
> Equipping artificial agents with useful exploration mechanisms remains a challenge to this day. Humans, on the other hand, seem to manage the trade-off between exploration and exploitation effortlessly. In the present article, we put forward the hypothesis that they accomplish this by making optimal use of limited computational resources. We study this hypothesis by meta-learning reinforcement learning algorithms that sacrifice performance for a shorter description length **(defined as the number of bits required to implement the given algorithm)**. The emerging class of models captures human exploration behavior better than previously considered approaches, such as Boltzmann exploration, upper confidence bound algorithms, and Thompson sampling. We additionally demonstrate that changing the description length in our class of models produces the intended effects: reducing description length captures the behavior of brain-lesioned patients while increasing it mirrors cognitive development during adolescence.
>
> >(Clarity) RL^3 seems like a poor method name given that RL^2 was so named since it was using reinforcement learning to train a reinforcement learning algorithm. Since this approach is not doing an extra meta-level of optimization, and is instead limiting the information in the weights of the inner RL algorithm, RL^3 is a very poor choice of name. What about RR-RL^2 for resource rational RL^2? Or IB-RL^2 for information bottleneck RL^2?
>
> While we like the name RL^3, we have to admit that it might cause confusion, as it does not add an extra meta-level as highlighted by the reviewer. We have therefore changed it, based on the reviewer’s suggestion, to RR-RL^2.

---

> > ### Author Response · Authors · 2022-08-01
> > **Reviewer 3 (Part 2)**
> >
> > >(Clarity) Figure 1a is somewhat difficult to interpet as a standalone figure. I think I understand it as behaving more like Thompson sampling as the description length is increased, but is that what’s actually happening? Some extra description in the figure caption would really help with the interpretation. (See also my question in the “questions” section below).
> >
> > We thank the reviewer for informing us that this figure could be improved by expanding on its description. We will add the following text to our camera-ready version to make the figure more self-contained:
> >
> > Illustration of exploration strategies implemented by RR-RL$^2$. (a) Probit regression coefficients obtained by fitting the hybrid model **(also see Equation 6 and the corresponding explanation in the main text)** to data simulated by RR-RL$^2$ with varying description lengths (depicted on a logarithmic scale). **The blue line shows the influence of the estimated mean on choices (Boltzmann exploration), the orange line shows the influence of the option’s uncertainty estimates (UCB-based exploration), and the green line shows the influence of the uncertainty-scaled estimated mean  (which corresponds to Thompson sampling in this particular task).** (b) UMAP embeddings of probit regression coefficients for RR-RL$^2$ and human participants.
> >
> > > (Clarity) Figure 2a - is RL^3 really at 0 BIC? Is that theoretically possible? Or is the delta BIC with respect to the RL^3 model? If the latter, the caption should be updated accordingly.
> >
> > The figure used to show the difference in BIC to the best-performing model (RR-RL$^2$). We agree that this can be confusing and have changed the figure to illustrate the actual BIC values instead (as also suggested by another reviewer). With this change, no update of the figure caption was necessary.
> >
> > >(Quality) Experimental results on development are much less compelling than for the other two experiments. The model is clearly doing something distinctly different from humans in that its base probability of exploration is 0 rather than a constant value. While the authors address some reasoning for why the effect size is lower for the model, there is no discussion of the differences in why the overall likelihood of exploration for people is so much higher. Could the authors comment on this?
> >
> > We thank the reviewer for this thoughtful comment. We will add the following paragraph to the camera-ready version to speculate on the differences in the base rate of random exploration:
> >
> > However, when comparing the effect sizes of our analysis to those from the experimental study, we find that the interaction effect between description length and horizon on directed exploration only amounts to around half of the effect between age and horizon. We speculated that part of this difference comes from a mismatch between the distribution used to train our models and what kind of tasks people expect in the experiment. People, for instance, might assume that task rewards are noisier than they are, which would require more exploratory choices, and, in turn, lead to stronger effects. We tested this hypothesis by retraining RR-RL$^2$ on the same distribution but with the standard deviation of the reward noise increased by $50\%$. While this modification increased the effect size of the interaction effect on directed exploration, it did not close the gap entirely, suggesting that there are additional -- currently undiscovered -- factors that contribute to the development of strategic directed exploration during adolescence. **Furthermore, we found that, while strategic random exploration was unaffected by both age and description length, humans had a higher base rate of engaging in random exploration than our models. We, however, believe that this issue could be resolved by adding an $\epsilon$-greedy error model to our models as doing so would naturally increase the base rate of random exploration.**

---

> > > ### Author Response · Authors · 2022-08-01
> > > **Reviewer 3 (Part 3)**
> > >
> > > >While reasonable, this is a rather limited view of both description length, as well as requiring fairly strong assumptions about the form of the decision-making process underlying explore/exploit. For example, the presented method could not obviously extend to consider object-oriented or model-based exploration, which is likely critical for more ecological/complex tasks than bandit settings. Indeed, much of the work in AI for exploration has focused on these concepts. How do the authors see this method as being applicable to these more complex settings?
> > >
> > > We agree with the reviewer that object-oriented and model-based exploration is crucial when moving to more complex tasks. Previous work has demonstrated that meta-learned algorithms approximate the Bayes-optimal policy, which is obtained by applying model-based planning to a larger belief-state MDP [1]. Furthermore, it has been shown that meta-learned algorithms similar to the one we have used solve reinforcement learning problems using model-based reasoning [2]. Thus, we think that it is possible for a meta-learned algorithm to implement such forms of exploration when trained on an appropriate task distribution.
> > >
> > > >The fit of RL^3 to both healthy adults and the lesioned population is very good, but the fit to the adolescent data is significantly worse. Both the effect size, but also the base rate of exploration, seem to be dramatically different from the human groups. Could the authors speculate on why this fit is not great, and whether there are changes to the model that could be made to improve the fit?
> > >
> > > We agree with the reviewer that this partial mismatch is perhaps the weakest point of the paper. However, while the model does not explain the developmental data fully, it still shows the key findings of the original study. We will also add an additional paragraph discussing the mismatch in the base rate of random exploration as described above to the camera-ready version.
> > >
> > > >One major missing piece for me was some intuition on how classic exploration strategies come out of the RL^3 approach. In particular, why is Thompson sampling seemingly the optimal strategy for exploration when a sufficiently high description length is allowed? And why does it require a high description length in the first place? Adding some intuition for this to the paper would help – especially in the first experiment for discussing Figure 1.
> > >
> > > We thank the reviewer for this comment. We will incorporate the reviewer’s suggestion in our camera-ready version and plan to provide the following intuition on what influences the strategies implemented in our models (inserted at the end of section 2):
> > >
> > > Which exploration strategies RR-RL$^2$ implements will partially depend on its available computational resources. Models with limited resources must implement strategies that rely on simple computations, such as computing average rewards or noisy estimates thereof. Models with access to more resources, on the other hand, can spend some of them to compute more complex statistics, such as uncertainty estimates, and incorporate these into their decision-making process. Moreover, what strategies are resource-rational not only depends on the computational resources of the decision-maker but also on the characteristics of the particular problem under consideration. Problems with a short task horizon, for instance, require less exploration and thereby make exploitation more appealing, while problems with a longer horizon allow for the application of more sophisticated exploration strategies.
> > >
> > > [1] Ortega, P.A., Wang, J.X., Rowland, M., Genewein, T., Kurth-Nelson, Z., Pascanu, R., Heess, N., Veness, J., Pritzel, A., Sprechmann, P. and Jayakumar, S.M., 2019. Meta-learning of sequential strategies. arXiv preprint arXiv:1905.03030.
> > >
> > > [2] Wang, J.X., Kurth-Nelson, Z., Tirumala, D., Soyer, H., Leibo, J.Z., Munos, R., Blundell, C., Kumaran, D. and Botvinick, M., 2016. Learning to reinforcement learn. arXiv preprint arXiv:1611.05763.

---

> > > > ### Comment · Reviewer_6vWe · 2022-08-07
> > > > **Thank you for the detailed reply!**
> > > >
> > > > Thank you for the detailed reply - I am very glad that the authors found the review helpful and I agree that the paper has been made substantially clearer. My concerns have been addressed. I would only ask the authors to update the PDF so that the title is the one they suggested: "Modeling human exploration through resource-rational reinforcement learning." and then I will be happy to increase my score.

---

> > > > > ### Author Response · Authors · 2022-08-07
> > > > > **We have changed the title**
> > > > >
> > > > > We have uploaded a new PDF and changed the title to "Modeling Human Exploration through
> > > > > Resource-Rational Reinforcement Learning". Thank you very much for all of your feedback!

---

> > > > > > ### Comment · Reviewer_6vWe · 2022-08-07
> > > > > > **I have raised my score accordingly**
> > > > > >
> > > > > > Thank you -- I have raised my score accordingly.

---

### Official Review · Reviewer_oRHY · 2022-07-12

**Rating:** 7
**Confidence:** 4
**Soundness:** 3 good
**Presentation:** 3 good
**Contribution:** 3 good

**Summary:**

The paper explores computational foundations underlying human exploration and focuses on the idea that model description length may be a critical factor determining the resulting type of exploration. The authors first summarise theoretical foundation of the model and then move on to comparing their proposed model (RL^3) to other models in their ability to fit behavioural data in normal participants as well as behavioural data in lesioned patients as well as changes with development. They show that their model outperforms other models regarding fit to behavioural data and that manipulating description length qualitatively reproduces the effect of brain lesions in Iowa Gambling Task as well as the effects of development in the Horizon task.

**Questions:**

For the first weakness, perhaps it would be better to link with more cognitive computational neuroscience papers (especially those linking working memory capacity to reinforcement learning, e.g. work of Collins and Frank), where biological correlates are a lot more straightforward, and at least provide discussion on plausible biological computation of quantities like KL.

For the second weakness, at the very least I would expect some discussion mentioning the possibility that simple models with dynamic control of exploration (and possibly other) metaparameters as opposed to full meta-learning with control of description length may be another feasible (and much simpler, even if not normative) solution to the problem of accurately reproducing behavioural data.

In Fig. 1 interpretation that "Between 100 and 1000 nats, we additionally find minor influences of a Boltzmann-like exploration strategy." I don't really understand what exactly this is based on? How do you define minor influences?

For Fig. 3 I would rather put data from D2 there as opposed to picking 2 "example models". If using choice probability as the only behavioural measure is already quite limiting, then picking 2 matching data points seems even more so, It's also curious that 'lesion group' performance seems to be better (at least in this one measure) than performance with any description length (<100) that best matches Boltzmann sampling. Again I'm very curious how a simple RL algorithm with dynamic metaparameter control or RL + working memory algorithms would perform...

**Limitations:**

Some of them yes, but some others not yet - see suggestions above in 'questions'.

**Strengths And Weaknesses:**

This paper discusses a really interesting hypothesis regarding what is the relevant factor determining the type and scale of exploration in the RL with metalearning set up. To my knowledge it is novel and it could provide some very stimulating ideas for the exploration-exploitation field. The authors also show great effort trying to link their results to behavioural (and lesion + developmental) studies and show a good understanding and review of relevant literature, both in machine learning and neuroscience.

In my opinion, weaknesses mainly fall into 2 categories:
- firstly, the level of methodological description is somewhat limited, and although in principle the suggested approach could be biologically plausible, it is hard to imagine (nor is there any discussion), how this could be biologically implemented.
- secondly, I feel that the benchmarks the authors' model is compared to are extreme / theoretical cases as opposed to using more empirical/heuristic ideas of how RL metaparameters (including those controlling exploration) should be controlled, e.g. Doya "Metalearning and neuromodulation" (Neural Networks 2002) or Luksys et al. 2009 Nat Neuroscience paper on modelling of stress, which incidentally also doesn't use bandit tasks (whose excessive use the authors rightly criticised). It may well be that while theoretical prototypes such as Boltzmann or Thompson may not do very well in detailed comparison with behavioural data, more empirical approaches, like those in the mentioned papers may do better, despite their simplicity and no need to compute complex quantities like KL.

---

> ### Author Response · Authors · 2022-08-01
> **Reviewer 2 (Part 1)**
>
> We thank the reviewer for the positive feedback. We have made several modifications to our text (as described below) and hope that these changes address the reviewer’s comments adequately.  Minor changes are already added to our newly uploaded version. For larger text changes, we describe what we will add to our camera-ready version (the camera-ready version allows for one additional page, but the rebuttal does not). Note that we have changed RL$^3$’s name to RR-RL$^2$ based on another reviewer’s request.
>
> > In my opinion, weaknesses mainly fall into 2 categories:
> >
> >Firstly, the level of methodological description is somewhat limited, and although in principle the suggested approach could be biologically plausible, it is hard to imagine (nor is there any discussion), how this could be biologically implemented.
>
> It is indeed an interesting question how our models could be implemented in biologically plausible ways. We will add the following sentences to the camera-ready version of our article to discuss this possibility:
>
> RR-RL$^2$ also places a constraint on a particular type of computational resource: the description length of the reinforcement learning algorithm in use. People, on the other hand, are subject to a variety of additional computational constraints. They can, for instance, only run algorithms with finite computation time or only store a restricted amount of chunks in their short-term memory. Future work should aim to unify all of these constraints in a common framework. **It would also be interesting to investigate how RR-RL$^2$ could be implemented in a biologically plausible way. We believe that replacing the standard recurrent neural networks used in this article with spiking neural networks could provide one promising path towards this goal. Earlier work of Bellec et al. [1] showed that it is indeed possible to train spiking neural networks in a meta-learning setting and would therefore provide an excellent starting point for this endeavor.**
>
> >Secondly, I feel that the benchmarks the authors' model is compared to are extreme / theoretical cases as opposed to using more empirical/heuristic ideas of how RL metaparameters (including those controlling exploration) should be controlled, e.g. Doya "Metalearning and neuromodulation" (Neural Networks 2002) or Luksys et al. 2009 Nat Neuroscience paper on modelling of stress, which incidentally also doesn't use bandit tasks (whose excessive use the authors rightly criticised). It may well be that while theoretical prototypes such as Boltzmann or Thompson may not do very well in detailed comparison with behavioural data, more empirical approaches, like those in the mentioned papers may do better, despite their simplicity and no need to compute complex quantities like KL.
>
> We think that models with dynamic hyperparameters, like the ones mentioned by the reviewer, are super interesting. In fact, the Thompson sampling model can be interpreted as a standard Q-learning model with dynamic hyperparameters. The total uncertainty (TU) acts as a temperature parameter as it scales the estimated value difference (V). This temperature parameter is dynamic as it changes based on how uncertain the agent is (the more uncertain the agent is, the nosier its choices). Because the updating for the reward distributions is given by the Kalman filtering equations, the model also has an adaptive learning rate. This learning rate is given by the Kalman gain, which is a function of the agent's uncertainty (see [2] for the corresponding equations). If the agent is more uncertain, it will learn faster. If it is already quite certain, it will learn slower. There are clearly many other ways how such hyperparameters can be controlled, and we believe that their scheduling could also be meta-learned as suggested by the reviewer. We discuss this possibility in our response to another issue below.
>
> [1] Bellec, G., Salaj, D., Subramoney, A., Legenstein, R. and Maass, W., 2018. Long short-term memory and learning-to-learn in networks of spiking neurons. Advances in neural information processing systems, 31.
>
> [2] Gershman, S.J., 2018. Deconstructing the human algorithms for exploration. Cognition, 173, pp.34-42.

---

> > ### Author Response · Authors · 2022-08-01
> > **Reviewer 2 (Part 2)**
> >
> > >For the first weakness, perhaps it would be better to link with more cognitive computational neuroscience papers (especially those linking working memory capacity to reinforcement learning, e.g. work of Collins and Frank), where biological correlates are a lot more straightforward, and at least provide discussion on plausible biological computation of quantities like KL.
> >
> > We thank the reviewer for pointing us towards the work of Collins and Frank. We have incorporated this reference into our updated manuscript (at the point where we discuss that our model could be extended to include limitations on other computational resources such as working memory).
> >
> > >For the second weakness, at the very least I would expect some discussion mentioning the possibility that simple models with dynamic control of exploration (and possibly other) metaparameters as opposed to full meta-learning with control of description length may be another feasible (and much simpler, even if not normative) solution to the problem of accurately reproducing behavioural data.
> >
> > We agree with the reviewer that testing other types of meta-learning, such as learning dynamic schedules for hyperparameters in standard reinforcement learning models, would be valuable. We are planning to add the following paragraph to the discussion of our camera-ready version to highlight some possibilities:
> >
> > Finally, there are many alternative ways of how one could apply meta-learning to our setting. It is, for example, possible to meta-learn hyperparameters of standard reinforcement learning algorithms \citep{doya2002metalearning, luksys2009stress}, or to use model-agnostic meta-learning to find optimal weight initializations for neural networks that are trained by gradient descent \citep{finn2017model}. Future work should compare these and other models against each other. For this to be effective, we believe that a set of commonly-accepted benchmarks needs to be established.
> >
> > >In Fig. 1 interpretation that "Between 100 and 1000 nats, we additionally find minor influences of a Boltzmann-like exploration strategy." I don't really understand what exactly this is based on? How do you define minor influences?
> >
> > We thank the reviewer for suggesting that we should be more precise with this statement. To do so, we have added the values of the average regression weights within these ranges to the text:
> >
> > We find that RL$^3$ implements a Boltzmann-like exploration strategy for description lengths between $1$ and $100$ nats. Note that behavior at this stage is quite noisy as indicated by the small probit regression weights **(average regression coefficient of $0.15$)**. Beginning from $100$ nats, we observe a rise of the factor corresponding to Thompson sampling, which continues to rise until the limit of $10000$ nats. Between $100$ and $1000$ nats, we additionally find minor influences of a Boltzmann-like exploration strategy **(average regression coefficient of $0.19$)**. For a description length of $1000$ nats and larger, Boltzmann-like exploration diminishes and is replaced with minor influences of a UCB-based strategy **(average regression coefficient of $0.23$)**.
> >
> > >For Fig. 3 I would rather put data from D2 there as opposed to picking 2 "example models". If using choice probability as the only behavioural measure is already quite limiting, then picking 2 matching data points seems even more so, It's also curious that 'lesion group' performance seems to be better (at least in this one measure) than performance with any description length (<100) that best matches Boltzmann sampling. Again I'm very curious how a simple RL algorithm with dynamic metaparameter control or RL + working memory algorithms would perform...
> >
> > We agree with the reviewer that Figure D2 is the more informative of the two. However, at the same time, we find Figure 3b more intuitive to understand and compare to Figure 3a. We, therefore, decided to keep the order as it was in our initial submission. We still refer the interested reader to the more detailed figure in Appendix D2 within the main text.

---

### Official Review · Reviewer_xjvi · 2022-07-17

**Rating:** 7
**Confidence:** 3
**Soundness:** 3 good
**Presentation:** 3 good
**Contribution:** 3 good

**Summary:**

The paper examines human exploration from the perspective of rationality under limited resources. The focus is on exploration in multi-armed bandit problems where the objective is to maximise the sum of rewards over a finite horizon. It is assumed that the agent has a prior distribution over bandit problems it may encounter.

The authors investigate the hypothesis that people explore optimally within the contraints they have on their computational resources. The authors devise a family of algorithms that achieve optimal exploration subject to a constraint on the description length of the algorithm. The authors then examine three data sets -- these are exploration data collected  from human subjects under different circumstances -- and demonstrate that their model captures certain characteristics of these data sets.


**Questions:**

When analyzing the data from Gershman (2018) -- the authors examined how well each model fitted the behavioral data from a two-armed bandit task -- my understanding is that C was a free parameter and that this would create a very flexible model that can fit many different types of exploration behaviour well. Would the authors agree? And would it be possible to examine the model in a predictive context? E.g., fit the parameters on some part of the data and examine how well the fitted model explains the remaining data?

When analyzing the data by Bechara et al. (1994) -- 100 choices in a single four-armed bandit problem -- the authors focused on the proportion of low- and high-risk arms selected across the entire experiment. The model (I believe) can make more precise predictions than that. Is a more rigorous test of the model possible and have the authors attempted that?



**Limitations:**

The main limitation is that all experimental data examined is on bandit problems. This is acknowledged by the authors.

**Strengths And Weaknesses:**

The paper explores a useful point of view and I found the results to be interesting and informative.

The model that is developed puts together ideas from various existing models and solution methods to create a specific approach to exploration under limited computational resources. The primary novelty in the paper is the analysis of human exploration data --  from very different experiments -- using this model.

I have some concerns about the flexibility of the model (and consequently, the interpretation of the results) and some of the design choices.

When analyzing the data from Gershman (2018) -- the authors examined how well each model fitted the behavioral data from a two-armed bandit task -- my understanding is that C was a free parameter and that this would create a very flexible model that can fit many different types of exploration behaviour well.

When analyzing the data by Bechara et al. (1994) -- 100 choices in a single four-armed bandit problem -- the authors focused on the proportion of low- and high-risk arms selected across the entire experiment. The model (I believe) can make more precise predictions than that and I wonder whether a more rigorous test of the model would have been possible.

The paper is generally clear, especially at high-level exposition of the motivations and ideas.

Additional  comments for the authors:

Figure 2a is misleading. The text in the  body of the  paper and the figure caption both note that the figure is showing Bayesian information criterion (BIC) values for the aggregated data. But the figure actually shows the difference in BIC compared to RL3, with RL3 value in the figure corresponding to a value of 0. In other  words, there is a mismatch in what the figure shows and what the text says about what the figure shows. Figure 2a somewhat misleading. The text in the  body of the  paper and the caption notes that the figure is showing Bayesian information criterion (BIC) values for the aggregated data. But the figure actually shows the difference in BIC compared to RL3  (with RL3 in the figure showing 0).

Corrections to some references:

"Hanna Damasio. Lesion analysis. Neuropsychology., 1989." The correct  reference is: Damasio, H., Damasio, A.R., 1989. Lesion Analysis in Neuropsychology. Oxford Univ. Press, Oxford.

"Gerd Gigerenzer and Peter M Todd. Simple heuristics that make us smart. Oxford University Press, USA, 1999."  The authors should be Gerd Gigerenzer and Peter M Todd, & the ABC Research Group.

---

> ### Author Response · Authors · 2022-08-01
> **Reviewer 1**
>
> We are glad that the reviewer found our work interesting and thank them for the helpful feedback. We discuss how we addressed the specific points raised by the reviewer below. These changes are already fully incorporated in our newly uploaded revision.
>
> > I have some concerns about the flexibility of the model (and consequently, the interpretation of the results) and some of the design choices.
> >
> > When analyzing the data from Gershman (2018) -- the authors examined how well each model fitted the behavioral data from a two-armed bandit task -- my understanding is that C was a free parameter and that this would create a very flexible model that can fit many different types of exploration behaviour well.
>
> The understanding that C is a free parameter is correct. It is also true that varying this parameter leads to a diverse set of exploration behaviors. However, we view this not as a problem for our analysis, but rather as a strength – by manipulating just a single parameter we capture much of the variance in the exploration strategies that people apply. Existing alternative models contain a similar number of free parameters and fit the data less well. We also rely on the Bayesian information criterion, which takes the number of free parameters into account when quantifying goodness-of-fit (see Appendix B for further details), in our main analysis.
>
> > When analyzing the data by Bechara et al. (1994) -- 100 choices in a single four-armed bandit problem -- the authors focused on the proportion of low- and high-risk arms selected across the entire experiment. The model (I believe) can make more precise predictions than that and I wonder whether a more rigorous test of the model would have been possible.
>
> We agree with the reviewer that our model could make more precise predictions beyond the proportion of low- and high-risk arms. However, the problem is that the data collected by Bechara et al. is not publicly available, and thus we are confined to verifying our model on the analyses reported in the original study.
>
> > Figure 2a is misleading. The text in the body of the paper and the figure caption both note that the figure is showing Bayesian information criterion (BIC) values for the aggregated data. But the figure actually shows the difference in BIC compared to RL3, with RL3 value in the figure corresponding to a value of 0. In other words, there is a mismatch in what the figure shows and what the text says about what the figure shows. Figure 2a somewhat misleading. The text in the body of the paper and the caption notes that the figure is showing Bayesian information criterion (BIC) values for the aggregated data. But the figure actually shows the difference in BIC compared to RL3 (with RL3 in the figure showing 0).
>
> We thank the reviewer for pointing out this inconsistency. We adjusted Figure 2a to show the actual BIC values instead of their differences.
>
> > Corrections to some references:
> >
> >"Hanna Damasio. Lesion analysis. Neuropsychology., 1989." The correct reference is: Damasio, H., Damasio, A.R., 1989. Lesion Analysis in Neuropsychology. Oxford Univ. Press, Oxford.
> >
> >"Gerd Gigerenzer and Peter M Todd. Simple heuristics that make us smart. Oxford University Press, USA, 1999." The authors should be Gerd Gigerenzer and Peter M Todd, & the ABC Research Group.
>
> We have fixed these references in our new version. Thanks for spotting them!
>
> > When analyzing the data from Gershman (2018) -- the authors examined how well each model fitted the behavioral data from a two-armed bandit task -- my understanding is that C was a free parameter and that this would create a very flexible model that can fit many different types of exploration behaviour well. Would the authors agree? And would it be possible to examine the model in a predictive context? E.g., fit the parameters on some part of the data and examine how well the fitted model explains the remaining data?
>
> We agree with the first point (also see our response above). Examining models in a predictive context is also a great suggestion. We have therefore added an additional model comparison that uses the AIC, which can be understood as an approximation to the idea outlined by the reviewer, to Appendix C.1. The results of this analysis are fully consistent with the results based on BIC reported in the main paper.
>
> > When analyzing the data by Bechara et al. (1994) -- 100 choices in a single four-armed bandit problem -- the authors focused on the proportion of low- and high-risk arms selected across the entire experiment. The model (I believe) can make more precise predictions than that. Is a more rigorous test of the model possible and have the authors attempted that?
>
> We agree that our model does make more precise predictions, but sadly a rigorous test of these predictions is not possible as the data from the original study is not publicly available (also see our response above).

---

### Meta-Review · Area_Chair_DGcr · 2022-08-26

**Recommendation:** Accept
**Confidence:** Certain

**Metareview:**

This paper presents a resource constrained variant of the RL^2 algorithm, and studies how well it models human exploration behavior in bandit tasks.  The resource constraint is the policy description length (in bits for the learned parameters).  Based on the resource constraint, the algorithm produces a mix of Boltzmann-like and Thompson-sampling exploration.  The space of exploration strategies from the algorithm is compared to human behavior data in three human populations.  The fit to human data is substantially better than many alternatives that have been considered in previous papers.

The reviewers found many strong contributions in this paper.  A particular strength was the ability of the proposed model to fit human behavior data from three substantially different populations in the past literature (xjvi, ). The reviewers also appreciated the clarity of the presentation that mixed ideas from neuroscience and machine learning (xjvi, orHY), and the motivation.  Reviewers raised multiple detailed questions about the approach and results.  The author response addressed each of the concerns in detail, and many reviewer suggestions were incorporated to improve the paper presentation.  No reviewers raised additional concerns after the author discussion, and remaining questions were limited to potential directions for future work.  Reviewers expressed interest in this work and in potential directions for future work that builds on the presented ideas.

Four reviewers indicate to accept this paper for its novel contribution of a better model of human exploration behavior in bandit tasks.  The paper is therefore accepted.

**Award:**

No

---

### Decision · Program_Chairs · 2022-09-14

Accept